# Effect of Drip Irrigation on Soil Water Balance and Water Use Efficiency of Maize in Northwest China

**Yahui Wang [1,2], Sien Li [1,2,\*], Yaokui Cui [3,\*], Shujing Qin [1,2], Hui Guo [1,2], Danni Yang [1,2] and Chunyu Wang [1,2]**

1 Center for Agricultural Water Research in China, China Agricultural University, Beijing 100083, China; yahuilan@163.com (Y.W.); qinshujing2010@126.com (S.Q.); guohui2018@cau.edu.cn (H.G.); 13366069828@163.com (D.Y.); wangchunyu205@163.com (C.W.)

2 Shiyanghe Experimental Station for Improving Water Use Efficiency in Agriculture, Ministry of Agriculture and Rural Affairs, Wuwei 733000, China

3 Institute of Remote Sensing and GIS, School of Earth and Space Sciences, Peking University, Beijing 100871, China

\* Correspondence: lisien@163.com or lisien@cau.edu.cn (S.L.); yaokuicui@pku.edu.cn (Y.C.); Tel.: +86-13811991479 (S.L.)

**Abstract:** Drip irrigation (DI) has been widely utilized for crops and its water-saving effect has been confirmed by numerous studies. However, whether this technology can save so much water under the field scale during practical application is still uncertain. In order to answer this question, evapotranspiration (ET), soil water content, transpiration and evaporation over the DI and border irrigation (BI) in an arid area of NW China were continuously measured by two eddy covariance systems, micro-lysimeters, the packaged stem sap flow gauges and CS616 sensors during 2014–2018 growing seasons. The results showed that the DI averagely increased crop water use efficiency (CWUE) by 11% per year against BI. The deep drainage under DI treatment was lower than BI by 8% averagely for the five-year period. While for the ET, the DI averagely decreased ET by 7% and 40mm per year against the traditional BI. The decrease in ET was mainly due to the significant reduction in soil evaporation instead of transpiration. Oppositely, we found that DI may increase maize (*Zea mays* L.) transpiration in some year for the better preponderant growth of crop. Thus, the accelerating effect on transpiration of DI and its reducing effect on soil evaporation should be considered simultaneously. In our experiment, DI only improved CWUE and WUE (water use efficiency) by 11% and 15% on average in a large farmland scale, unable to always be more than a 20% improvement, as concluded by many other field experiments. Consequently, the water-saving effect of DI should not be overestimated in water resource evaluation.

**Keywords:** water use efficiency; border irrigation; drip irrigation; soil water balance

## 1. Introduction

Water scarcity is becoming more and more serious globally as a result of climate change and population increase [1]. How to maximize the use of water in irrigated agriculture has been the focus of many researchers. Film-mulched drip irrigation technology is a kind of new surface irrigation technology to meet the water-saving agriculture development and has been widely considered as a reliable irrigation way in terms of water saving and production increasing, and thus widely promoted and used in arid and semi-arid regions. In this irrigation method, the drip irrigation lines are laid on the soil surface and under the plastic film, and the water enters the soil surface through the drip irrigation line's emitters and gradually wets the soil. Due to the advantages of mulching and drip irrigation, when compared with traditional extensive irrigation, many studies have proved that drip irrigation under film can save water, increase temperature, promote crop growth, raise yield and improve water and fertilizer utilization efficiency [2–7].

To date, many researchers have investigated the effect of drip irrigation on water use efficiency. Verma (2007) [8] conducted experiments on a peach field, finding out film

mulching drip irrigation saved irrigation water by 59.1%, and promoted fruit yield by 30.4% against surface irrigation. Fan (2012) [9] reported film mulching drip irrigation could save 41.9% of water, increase 23.2% of cotton yield, and improve WUE by 70.7%, compared with furrow irrigation. Zhang (2016) [10] indicated drip irrigation under film mulch increased WUE of oil sunflower by 47.48%, 11.18% and 31.26%, respectively, compared with that of drip irrigation, film mulching sprinkling irrigation and furrow irrigation under mulching. Zhang (2017a) [11] studied different plastic films under drip irrigation in potato and confirmed that the film mulch drip irrigation averagely increased WUE by 9% during 2014–2015. Zhang (2018) [12] indicated drip irrigation under mulch reduced 2.8–5.2% evapotranspiration and enhanced WUE by 10.7–13.1% in the maize field for 3 years. Liao (2019) [13] used the method of water balance to conclude that drip irrigation under film mulch could averagely enhance WUE by 19% and decrease ET by 13–24% for 3 years compared with furrow irrigation in a cherry field. Han (2019) [14] conducted a 3-year field experiment in a cotton field indicating that drip irrigation under mulch significantly decreased the ratio of soil evaporation to evapotranspiration by 9%.

Based on the previous studies, we can conclude that drip irrigation can improve crop WUE against the traditional irrigation method by more than 20%, reduce evapotranspiration by more than 15% and influence the process of water balance. However, these results were mainly from the contrast experiments using the traditional measurement methods at a small zone scale, with representativeness of the results to be verified. If applied in a large area, such as 1000 hectare, whether this technology can save so much water will still be uncertain and doubtful. Consequently, answering the question is so critical for extending the technology in arid regions. Our experimental region faces the severest water scarcity in NW China due to insufficient rainfall, excessive evaporation, and uneven distribution of rainfall during the year. Irrigation under plastic mulch is the commonest farmland management of maize in the region. In recent years, to meet the demand from water-saving agriculture and relieve the increasingly severe regional water consumption crisis, DI was widely introduced and popularized.

For that end, we spent a lot conducting a long-term flux contrast experiment between the drip-irrigated maize fields and the border-irrigated maize fields of large area in northwest China during 2014–2018. Evapotranspiration, soil water content, transpiration and evaporation over drip irrigation (DI) and border irrigation (BI) treatments were continuously measured by two eddy covariance (EC) systems, micro-lysimeters, the packaged stem sap flow gauges and CS616 sensors, to reveal the water-saving effect of DI in a regional scale by contrast measurements and analysis. Detailed long-term field observations obtained through five-year experiments help us better understand the quantitative impact of large-scale DI in arid areas on farmland water balance and WUEs.

## 2. Materials and Methods

### 2.1. Experimental Site

The experimental sites were located in Wuwei City in Gansu Province in northwest China. Field observation was carried out for 5 years (2014–2018) at the Shiyanghe Experimental Station (37°52′ N, 102°50′ E, elevation 1581 m). The studied area has a typical temperate continental arid climate, which is characterized by 8 °C as an annual mean temperature, about 3550 °C as an annual accumulated temperature (>0 °C), approximately 2000 mm as a mean annual evaporation, 164 mm as an annual precipitation, 3000 h as an average annual sunshine duration and about 40 to 50 m as groundwater table [15–19]. The soil texture is generally silty loam at 100 cm depth. The most of precipitation in the region happens from July to September in summer.

### 2.2. Experimental Design

The contrast experiments were conducted during 2014–2018 on maize: film mulching drip irrigation (DI) treatment and film mulching border irrigation (BI) treatment. The drip irrigation Emitters with 3.2 L h$^{-1}$ discharge rate were every 0.3 m along the drip

line. The drip irrigation system (Dayu Irrigation Group Co., Ltd.; Wuwei, China) are planned by the government. The drip irrigation system includes water sources, deep pumps (40 kw), centrifugal filter, pressure pump (20 kw), laminated filter, capillary (Φ 16 drip line), branch (Φ 32 PE), trunk (Φ 63 PVC), pressure differential fertilization tank (30 L), mesh filter, water meter, pressure gauge and other accessories. The border irrigation system are planned by the farmers. It includes water sources, deep pumps (40 kw), branch (Φ 32 PE), trunk (Φ 63 PVC), water meter, pressure gauge and other accessories. Field management measures (Irrigation and fertilization) of both treatments conformed to the local farmers' traditional management practice.

BI is the main irrigation management method for maize by local farmers. The BI treatment covered an area of 400 m × 200 m from 2014 to 2015 and an area of 500 m × 250 m from 2016 to 2018. The 100cm of soil was characterized by 1.52 g cm$^{-3}$ as a mean soil dry bulk density and 0.29 cm$^3$ cm$^{-3}$ as a field capacity during 2014–2015. Moreover, the 100 cm of soil was characterized by 1.52 g cm$^{-3}$ as a mean soil dry bulk density and 0.32 cm$^3$ cm$^{-3}$ as a field capacity during 2016–2018. However, film mulching drip irrigation is a water-saving irrigation method promoted locally in recent years. The area of DI treatment was 2000 m×1000 m in 2014 and 2015, and 400 m × 200 m during 2016–2018. The mean soil dry bulk density was 1.52 g cm$^{-3}$ and field capacity 0.30 cm$^3$ cm$^{-3}$ in 2014 and 2015. In addition, the 100 cm soil had the mean soil dry bulk density 1.52 g cm$^{-3}$ and soil field capacity 0.29 cm$^3$ cm$^{-3}$ from 2016 to 2018. The experimental sites were shown in detail in previous study of Wang [20].

### 2.3. Methods and Measurements

In each treatment, an EC system was installed. The EC system of BI treatment in 2014 included a 3D sonic anemometer/thermometer, a Krypton hygrometer, a temperature and humidity sensor, a net radiometer and two soil heat flux plates, as described by previous studies [15–18]. During 2015–2018 under BI treatment, the new EC system consisted of a $CO_2$/$H_2O$ open path gas analyzer, two temperature and humidity probes, a Kipp & Zonen radiometer, two soil heat flux plates, five CS616 probes, five soil thermocouple probes, and an infrared radiometer. Five water content sensors and five soil temperature probes were set at every 20 cm of 100 cm depth, respectively. The EC system under DI treatment during 2014–2018 was same as that under BI treatment from 2015–2018. The energy flux data were measured every 30 min with the EC system and the sampling frequency was 20 Hz. These instruments and data processing methods have been detailed described by previous studies of Li (2018) and Wang (2020) [19,20].

Maize transpiration was measured with Stem-flow gauges (Flow32-1K, Dynamax Co., USA). Each device has eight probes to be installed on maize stems about 20 cm above the ground. Then we used the suitable calculation method for seed maize described in detail by Jiang (2014) [21] to caculate the transpiration rate of the crop stems. Soil evaporation was obtained on a daily basis by micro-lysimeters. The micro-lysimeter PVC tubes were 20 cm at height and 10 cm in diameter, respectively. Previous observations by Qin (2018) [22] indicated evaporation under mulch was considered non-negligible. The micro-lysimeters were buried in the middle of mulch (three replications) and the bare area between mulches (three replications), respectively. The irrigation amount was measured by water meters. These instruments and detailed data processing methods have been described by Wang (2020) [20].

The drainage amount (*D*) was obtained by the water balance equation [23,24]:

$$D = P + I + \Delta W + C - E - T - R \tag{1}$$

where *D* is drainage under the observation depth (mm), *E* evaporation (mm), *T* transpiration (mm), *P* precipitation (mm), *I* irrigation amount (mm), $\Delta W$ change in soil water storage (mm), *C* capillary rise (mm) and *R* runoff (mm). Meanwhile, as the field was flat and the groundwater table was 30–40 m, the value of both *R* and *C* was zero.

### 2.4. Data Interpolation

The experimental methods of evaporation (*E*) and transpiration (*T*) in our study missed the data in the days with the big winds, heavy rain or irrigation events. To analyze the variation of T and E during the whole growing period. We used the adjusted Shuttleworth–Wallace model introduced by Li (2013b) and Qin (2018) [16,22] to interpolate the missing data. The simulated values have a good agreement with the measured values (Figure 1).

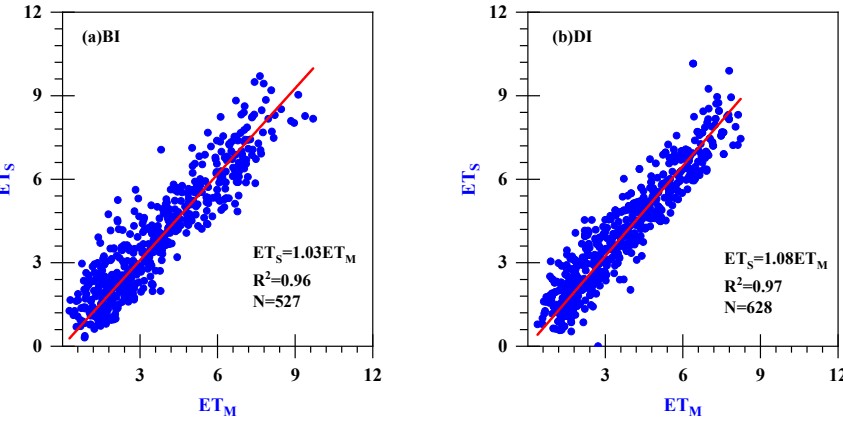

**Figure 1.** Comparison of simulated value of evapotranspiration ($ET_S$) by adjusted Shuttleworth-Wallace model and measured value of evapotranspiration ($ET_M$) over the maize field from 2014 to 2018 under film mulching border irrigation (BI) treatment and film mulching drip irrigation (DI) treatment.

### 2.5. Data Analysis Method

Two water use efficiencies were used in our study. We have referred to the calibration of crop water use efficiency (*CWUE*) in previous study [25]. However, evaporation always was considered as the ineffective component of water use, and transpiration related to crop growth. Considering that the CWUE was obtained as follows:

$$CWUE = \frac{T}{I + P} \qquad (2)$$

The water use efficiency (*WUE*, kg m$^{-3}$) was obtained as Kijne (2003) [26]. Some researchers name it "Crop Water Productivity". It is as follows:

$$WUE = \frac{Y}{E + T} \qquad (3)$$

where *Y* represents the maize yield (t hm$^{-2}$).

The Irrigation water use efficiency (*IWUE*, kg m$^{-3}$) was referred to Knox (2013) [27]. The calibration is as follows:

$$IWUE = \frac{Y}{I} \qquad (4)$$

In this paper, SPSS software was used to conduct *t*-test for components of soil water balance and WUEs in the drip irrigation and border irrigation, respectivly, and to analyze whether the differences between the data were significant or not.

## 3. Results

### 3.1. Soil Water Balance

3.1.1. The Difference of Precipitation (P), Irrigation (I) and Soil Water Content (SWC), Drainage (D) between DI Treatment and BI Treatment

As for the irrigation, the total irrigation amount under BI treatment was 360 mm, 550 mm, 480 mm, 570 mm, and 525 mm in 2014, 2015, 2016, 2017 and 2018, respectively,

while this figure under DI treatment was 350 mm, 400 mm, 427 mm, 368 mm, and 422 mm over the same period, respectively. Compared to the BI treatment, the DI treatment reduced irrigation amount by 10 mm, 150 mm, 53 mm, 202 mm and 103 mm, respectively, during this period. On average, drip irrigation reduced the irrigation amount of maize by 104 mm per year during the five years (Figure 2 and Table 1).

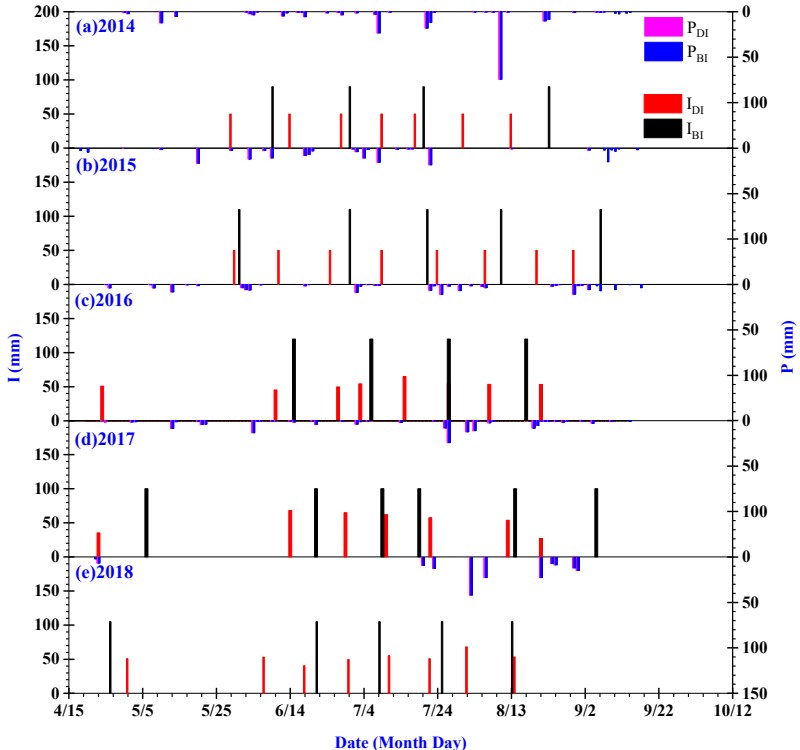

**Figure 2.** Seasonal variations of precipitation (P) and irrigation (I) over the maize field from 2014 to 2018 under film mulching border irrigation (BI) treatment and film mulching drip irrigation (DI) treatment.

**Table 1.** Water balance components, D/(P+I), T/(P+I) (CWUE), Y/I (IWUE) and Y/ET (WUE) under film mulching border irrigation (BI) and film mulching drip irrigation (DI) during 2014–2018 and *p* value by *t*-test of these.

| Treatment | Year | Days | ET (mm) | E (mm) | T (mm) | P (mm) | I (mm) | ΔW (mm) | D (mm) | Yield (t/hm²) | D/(P+I) | T/(P+I) | Y/I (kg/m³) | Y/ET (kg/m³) |
|---|---|---|---|---|---|---|---|---|---|---|---|---|---|---|
| BI | 2014 | 149 | 497 | 108 | 389 | 201 | 360 | −31 | 95 | 6.9 | 17% | 69% | 1.92 | 1.39 |
| DI | 2014 | 134 | 479 | 108 | 371 | 195 | 350 | −11 | 78 | 9.04 | 14% | 68% | 2.59 | 1.89 |
| BI | 2015 | 155 | 616 | 190 | 426 | 151 | 550 | −8 | 93 | 8.52 | 13% | 61% | 1.55 | 1.38 |
| DI | 2015 | 132 | 517 | 95 | 421 | 119 | 400 | −8 | 10 | 9.97 | 2% | 81% | 2.49 | 1.93 |
| BI | 2016 | 154 | 521 | 106 | 415 | 119 | 480 | −47 | 124 | 10.44 | 21% | 69% | 2.17 | 2.00 |
| DI | 2016 | 144 | 511 | 99 | 412 | 115 | 427 | −1 | 32 | 10.95 | 6% | 76% | 2.56 | 2.14 |
| BI | 2017 | 147 | 581 | 122 | 459 | 133 | 570 | 50 | 72 | 6.76 | 10% | 65% | 1.18 | 1.16 |
| DI | 2017 | 142 | 490 | 95 | 394 | 134 | 368 | −48 | 61 | 5.29 | 12% | 79% | 1.44 | 1.08 |
| BI | 2018 | 159 | 525 | 117 | 408 | 158 | 525 | 56 | 103 | 7.53 | 15% | 60% | 1.44 | 1.44 |
| DI | 2018 | 146 | 543 | 97 | 446 | 156 | 422 | 29 | 6 | 7.92 | 1% | 77% | 1.88 | 1.46 |
| BI | 2014–2018 | 153 | 548 | 129 | 419 | 152 | 497 | 4 | 97 | 8.03 | 15% | 65% | 1.62 | 1.47 |
| DI | 2014–2018 | 140 | 508 | 99 | 409 | 144 | 393 | −8 | 37 | 8.63 | 7% | 76% | 2.20 | 1.70 |
| *p*-value (*t*-test) | | | 0.074 | 0.043 | 0.47 | 0.002 | | 0.564 | 0.001 | 0.548 | 0.011 | 0.001 | 0.043 | 0.267 |

The daily variation of precipitation (P) and irrigation (I) under BI treatment and DI treatment during 2014–2018 is shown in Figure 2. During 2014–2018, drip irrigation reduced the precipitation of maize by an average of 8 mm (5%) for the shorter growth stages of DI treatment.

The daily variations of 0–100 cm SWC under BI treatment and DI treatment during 2014–2018 are shown in Figure 3. Due to more irrigation frequencies under DI treatment,

the fluctuation time of SWC in the whole growth period was more than that under BI treatment, but the fluctuation range of SWC under DI treatment was less than that under BI treatment during 2014–2018. The absolute change of SWC (ΔW) under BI treatment was 31 mm, 8 mm, 47 mm, 50 mm, and 56 mm in 2014, 2015, 2016, 2017 and 2018, respectively, while this figure under DI treatment was 11 mm, 8 mm, 1 mm, 48 mm, and 29 mm in the five years, respectively. Compared to the BI treatment, the DI treatment saved ΔW by 20 mm, 0 mm, 46 mm, 2 mm and 27 mm, respectively, in the five years. By comparison, drip irrigation reduced the absolute ΔW of maize by 18 mm averagely during 2014–2018.

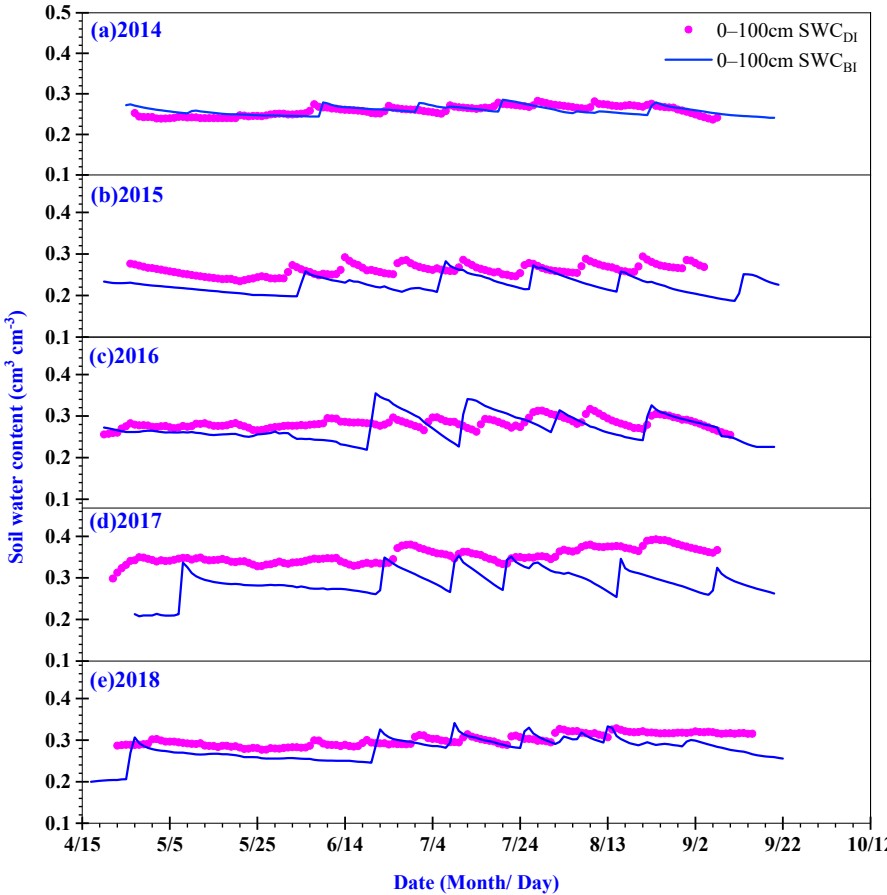

**Figure 3.** Seasonal variations of average soil water content of 100 cm depth measured by CS616 probes during the maize growing season from 2014 to 2018 under film mulching border irrigation (BI) treatment and film mulching drip irrigation (DI) treatment.

The total drainage under BI treatment was 95 mm, 93 mm, 124 mm, 72 mm and 103 mm in 2014, 2015, 2016, 2017 and 2018, respectively, while this figure under DI treatment was 78 mm, 10 mm, 32 mm, 61 mm, and 6 mm in the five years, respectively. Compared to the BI treatment, the DI treatment reduced drainage by 17 mm, 83 mm, 92 mm, 12 mm and 96 mm, respectively, in the five years. On average, drip irrigation markedly reduced the drainage of maize field by 60 mm per year (62%) (Table 1 and Figure 4).

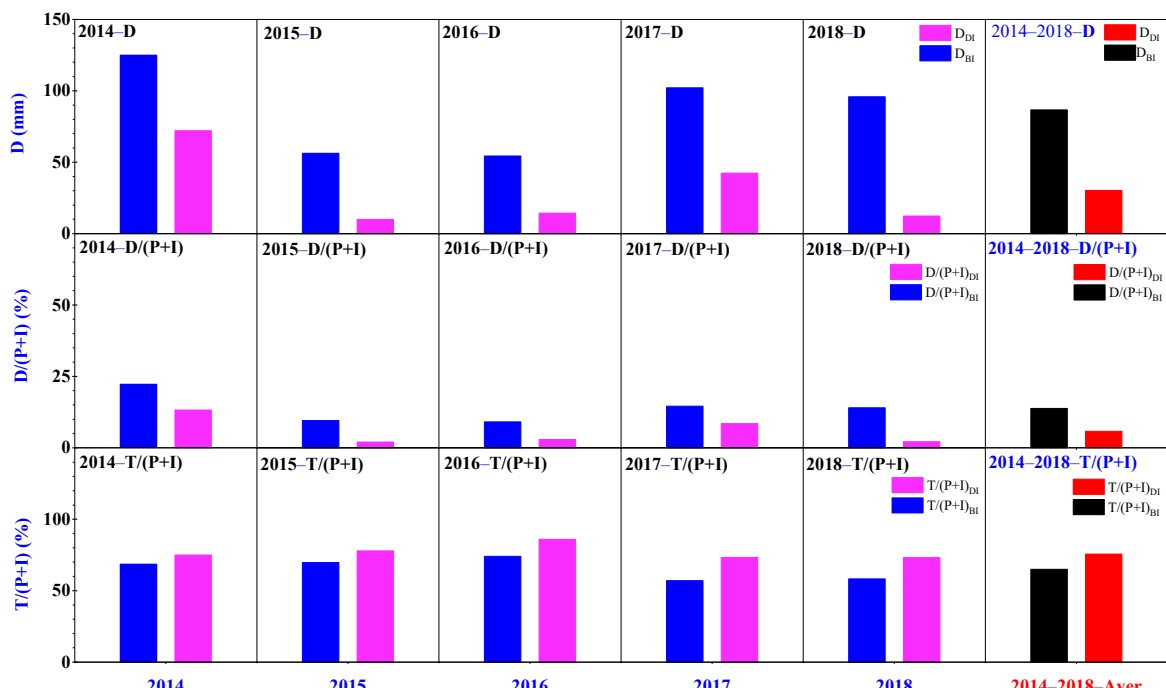

**Figure 4.** Comparison of drainage (D), D/(P+I) and effective water utilization (T/(P+I)) under BI treatment and DI treatment during 2014–2018.

In conclusion, DI treatment significantly decreased water input and output, which is mainly due to reduction on irrigation and drainage by 21% and 62% in an arid region. In addition, DI made the field water storage in a relatively stable state, which is beneficial to provide the water conditions needed for the growth of crops.

### 3.1.2. The Difference of Evaporation (E), Transpiration (T) between DI Treatment and BI Treatment

The daily variations of transpiration (T) and evaporation (E) under BI and DI treatments during 2014–2018 are shown in Figure 5. The total E under BI and DI treatments was 108 mm and 108 mm in 2014, 190 mm and 95 mm in 2015, 106 mm and 99 mm in 2016, 122 mm and 95 mm in 2017, and 117 mm and 97 mm in 2018, respectively. The DI treatment reduced evaporation by 0 mm, 94 mm, 8 mm, 27 mm and 20 mm, respectively, over 2014–2018, and averagely lowered evaporation by 30mm per year against BI treatment.

As for the transpiration, the total T under BI treatment and DI treatment was 389 mm and 371 mm in 2014, 426 mm and 421 mm in 2015, 415 mm and 412 mm in 2016, 459 mm and 394 mm in 2017, and 408 mm and 446 mm in 2018, respectively. Transpiration under DI treatment was saved by 18 mm, 5 mm, 3 mm, 65 mm and −38 mm, respectively, against BI treatment in the five years. Thus, drip irrigation may decrease or increase crop transpiration.

As for the five-year period, drip irrigation only reduced transpiration by 10 mm and less than 2.5% ET per year averagely. Thus, we could infer that the water-saving effect of drip irrigation was mainly due to reduction in soil evaporation rather than crop transpiration. In the opposite, drip irrigation may accelerate crop water demand and consumption. This is a very important finding, which will be discussed in the later section.

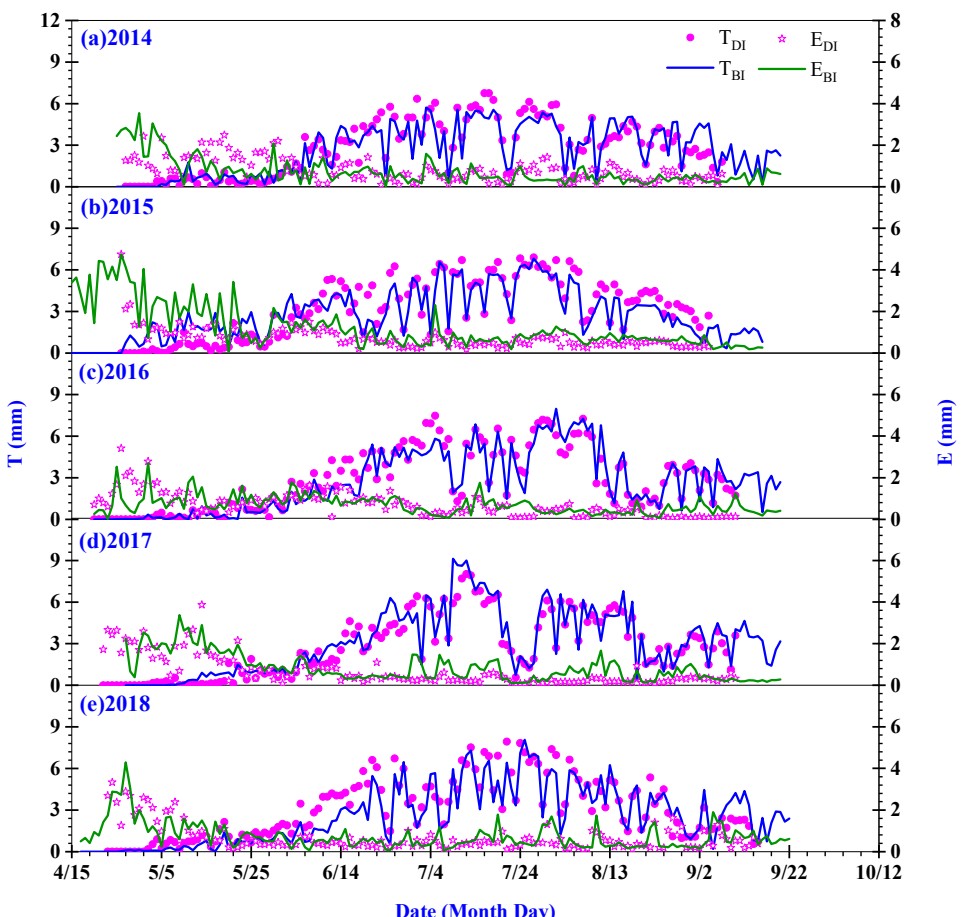

**Figure 5.** Seasonal variations of daily transpiration (T) measured by sap flow and daily evaporation (E) measured by micro-lysimeters under film mulching border irrigation (BI) treatment and film mulching drip irrigation (DI) treatment during 2014–2018.

*3.2. Comparison of Proportion of CWUE, WUE, IWUE and Proportion of Drainage to the Sum of Precipitation and Irrigation (D/(P+I)) between DI Treatment and BI Treatment*

The annual T/(P+I) (CWUE) and D/(P+I) under the BI and DI treatments during 2014–2018 are shown in Figures 4 and 6 and Table 1. The IWUE under BI treatment was 69%, 61%,69%, 65% and 60% in 2014, 2015, 2016, 2017, and 2018, respectively, while this figure under DI treatment was 68%, 81%, 76%, 79%, and 77% in the five years, respectively. Compared to the BI treatment, the DI treatment increased CWUE by −1%, 20%, 7%, 13% and 17% in the five years. By contrast, film mulching drip irrigation reduced CWUE of maize by an average of 11%.

While the WUE under BI treatment was 1.39 kg m$^{-3}$, 1.38 kg m$^{-3}$, 2.00 kg m$^{-3}$, 1.16 kg m$^{-3}$, and 1.44 kg m$^{-3}$ in 2014, 2015, 2016, 2017, and 2018, respectively (Table 1), while this figure under DI treatment was 1.89 kg m$^{-3}$, 1.93 kg m$^{-3}$, 2.14 kg m$^{-3}$, 1.08 kg m$^{-3}$, and 1.46 kg m$^{-3}$ in the five years, respectively. Compared to the BI treatment, the DI treatment promoted WUE by 36%, 39%, 7%, −7% and 2% in the five years. By contrast, film mulching drip irrigation enhanced WUE of maize by an average of 15%. The DI treatment enhanced IWUE by 35%, 61%, 18%, 22% and 31% in the five years, this is a very significant improvement.

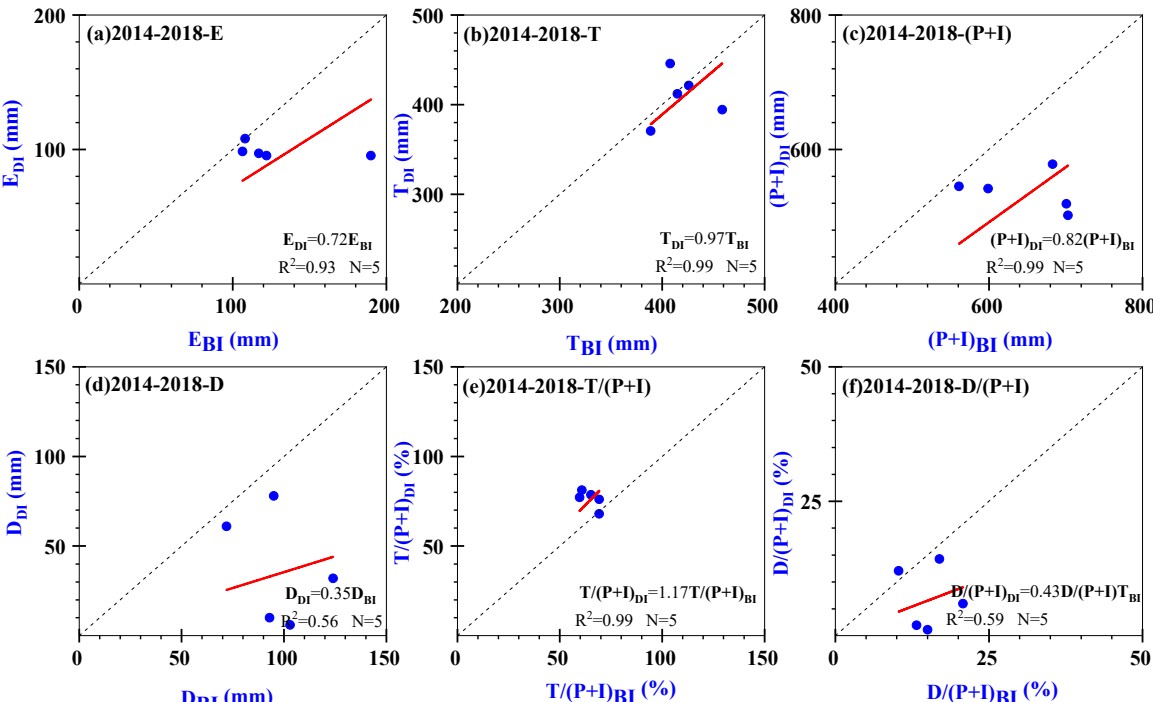

**Figure 6.** Comparison of evaporation (E), transpiration (T), drainage (D), P+I, D/(P+I) and effective water utilization (T/(P+I)) under film mulching border irrigation (BI) treatment film mulching drip irrigation (DI) treatment during 2014–2018.

As for D/(P+I), the proportion of drainage to the sum of precipitation and irrigation (D/(P+I)) under BI treatment was 17%, 13%, 21%, 10%, and 15% in 2014, 2015, 2016, 2017 and 2018, respectively, while this figure under DI treatment was 14%, 2%, 6%, 12%, and 1% in the five years, respectively. Compared to the BI treatment, the DI treatment increased D/(P+I) by 3%, 11%, 15%, −2% and 14% in the five years. By contrast, drip irrigation reduced the D/(P+I) of maize by an average of 8% per year.

The results showed that DI significantly improved the WUE and CWUE. This result further verified the water-saving effect and feasibility of DI in arid areas. Many studies have shown that the amount of leakage increases with the amount of irrigation, but our study found that drip irrigation could reduce this increase rate (D/(P+I)). This finding also provides theoretical support for the development and application of drip irrigation technology.

## 4. Discussion

### 4.1. Why Does Drip Irrigation Improve Water Use Efficiencies?

Water resource shortage is one of the major characteristics of arid regions, where development of water-saving agriculture is the concern of many scholars. Three different WUE indexes were selected to comprehensively evaluate the performance of DI in WUE from the aspects of economic production benefit (WUE and IWUE), crop water consumption and component of water consumption (CWUE) [25–28]. Meanwhile, multiple indexes were selected to analyze the influence of drip irrigation on water use efficiency in order to provide data reference and theoretical support for future related studies. Our results showed that the change of irrigation methods from BI to DI could improve the maize CWUE, WUE and IWUE by 11%, 15% and 36% per annum during 2014–2018. This indicates that DI has a greater water-saving capacity than BI in arid area, and the application of DI technology is also an effective measure to improve the WUE of farmland [29].

Two mainly factors contributed to the increase on WUE of DI. One is the significant decrease of water input under DI, which is mainly due to reduction on irrigation by 21% under DI in our study. With little amount and high frequency of DI, the soil moisture

content of the root layer maintained a relatively stable value for a long time. At the same time, little water infiltrated in the deep soil layer, thus utilizing more irrigation water. The irrigation pipes of DI were laid under the film, which makes the film mulching increase the evaporation resistance, and thus changes the vapor exchange of surface water [30]. The other factor is that the DI promotes the crop growth. Moreover, the superior growth of the crop contributed to the increase of maize yield (Table 1). Transpiration is closely related to the growth of plant. Our study showed the annual transpiration under two treatments was similar and concluded that more water-saving DI would not reduce transpiration of crops, but even enhance transpiration to promote the growth of crops due to the warming effect of DI. This is also consistent with previous studies showing that DI accelerates crop growth [7,31–34]. Many previous studies have shown that DI can improve utilization efficiency of irrigation water [6,35]. The research of Geerts and Raes have considered DI aimed to maximize water productivity and to stabilize—rather than maximize—yields [36].

### 4.2. Compared with Precious Studies?

Our Research analyzed differences in farmland water balance and WUE between DI and BI in arid areas of northwest China. The research shows that DI has a significant improvement in water-saving capacity in drought-stricken areas compared with traditional BI. Many previous studies have focused on the development of irrigation systems when implementing DI techniques on many crops, and different methods of implementing the same irrigation method will result in different WUE and yield [37–39]. Different irrigation systems and management methods under different DI conditions in different crops and regions can increase the WUE and yield by more than 15% [36,40,41], which indicates that DI has strong water-saving potential and needs to be developed urgently. Our study showed that DI improved WUE by 15% compared with BI without irrigation optimization. However, how much water-saving space can be improved by implementing DI with optimized irrigation system in a large area should be the direction of our further research.

Our research shows that the DI reduced the drainage and D/(P+I) by an average of 62% and an average of 8% in maize fields. This is a very significant improvement. The main reason is that DI could reduce per-time irrigation norm and therefore shallowed the wetting front depth and reduced the wetting area of irrigation water in the soil, thus resulting in reduction of water going deep into the soil, to effectively reduce deep drainage loss of water [42–44]. Second, DI increases the irrigation frequency. The SWC under film-mulched irrigation had a low fluctuation range during the whole growth period. Compared with traditional irrigation, the water losses of both the upper (soil evaporation) and lower (deep drainage) borders are lower. It has shown that the water leakage increases with the amount of irrigation. Meanwhile, our study found that DI can reduce this increase rate (D/(P+I)). This finding also provides theoretical support for the development and application of drip irrigation technology. Meanwhile, DI reduces the loss of water leakage, which may avoid problems such as groundwater pollution, but in arid areas where evaporation is greater, salt and fertilizer may remain more on the soil surface, causing soil compaction and other problems. However, how to avoid these problems by regulating DI fertilization and irrigation system still needs further research.

### 5. Conclusions

Having conducted a five-year continuous contrast observation field experiment under border irrigation and drip irrigation treatments, we concluded that drip irrigation averagely increased maize CWUE and WUE by 11% and 15%, decreased ET by 7% and 40mm per year against the traditional border irrigation. The decrease in ET was mainly due to the significant reduction in soil evaporation rather than transpiration. In the opposite, drip irrigation may increase maize transpiration in some year for the better preponderant growth of crop. The increasing effect on transpiration of drip irrigation and its reducing effect on soil evaporation should be considered simultaneously. Furthermore, our research revealed that the drip irrigation can only improve CWUE and WUE by 11% and 15%

averagely in a farmland scale, unable to always over 20% as concluded by many field experiments. Thus the water-saving effect of drip irrigation shouldn't be overestimated in a large regional scale.

Our research revealed the water-saving mechanism of drip irrigation, and evaluated the water-saving effect of technology in regional scale. These provided critical scientific basis for understanding and promoting the drip irrigation technology. However, the study focused only on the impact of changes in border irrigation conversion to drip irrigation on water use efficiency, and failed to explore the water-saving space generated by better irrigation systems under drip irrigation conditions. In addition, the beneficial and harmful effects of drip irrigation on farmland ecosystem still need further research and exploration. These are the directions that we need to pay attention to in the future.

**Author Contributions:** Conceptualization, Y.W. and S.L.; Methodology, Y.W., S.L. and Y.C.; Validation, Y.W., S.L. and Y.C.; Formal Analysis, Y.W.; Investigation, Y.W., S.Q., H.G., D.Y. and C.W.; Resources, S.L.; Writing—original draft preparation, Y.W.; Writing—review and editing, Y.W. and S.L. Project administration, S.L.; Funding acquisition, S.L. All authors have read and agreed to the published version of the manuscript.

**Funding:** This research was funded by Chinese National Natural Science Fund (51622907, 51879262, Grant No. 41901348) and the Key R&D Program of the Ministry of Science and Technology, China (Grant No. 2017YFE0122400).

**Institutional Review Board Statement:** Not applicable.

**Informed Consent Statement:** Not applicable.

**Data Availability Statement:** No new data were created or analyzed in this study. Data sharing is not applicable to this article.

**Acknowledgments:** We wish to thank the staff of the Shiyang River Basin Experimental Station for Agriculture and Ecological Water Saving of China Agriculture University.

**Conflicts of Interest:** The authors declare no conflict of interest.

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
