# Peer review of "Effect of Drip Irrigation on Soil Water Balance and Water Use Efficiency of Maize in Northwest China"

_water, doi:10.3390/w13020217_

Round 1

Reviewer 1 Report

Comments to the Authors

The entitled manuscript of Effect of Drip Irrigation on Soil Water Balance and Water Use

Efficiency of Maize in Northwest China provides valuable and substantial information on the comparison between DI and BI efficiency indices for maize in Northwest China. Below my specific comments and suggestions that need to implement into this manuscript to be publishable.    

Specific Comments and Suggestions

Abstract:

Overall, I did not notice that the authors mentioned the crop name that they conducted their experiment on, regardless of the one mentioned on line 24. When the authors mention the crop's name, they need to provide the scientific name where its first mentioned in the abstract and the introduction sections.

 I did not notice that the authors defined these acronyms: CWUE, ET, WUE, and BI? So how come the reader will understand what are these acronyms stand for?  The authors must define these acronyms in the abstract and do the same for the introduction section, where is mentioned first.

Line 16-19: where this experiment was conducted?

Line 17: what are these two treatments?

Line 19: revise to…during 2014-2018 growing seasons.

Line 19: The authors should define DI. It must be done at the first mention of the term drip irrigation. (i.e., Line 14: Drip irrigation (DI) has been widely….).

Introduction:

Generally, this section is short and needs more elaboration (i.e., cited works). The authors may also identify the type of drip irrigation cited from others' works (if applicable); was it surface drip irrigation or subsurface drip irrigation? There are significant differences between the two types. Therefore, it is substantial to write a comprehensive introduction that enables the reader to understand this irrigation system.    

Materials and Methods:

Line 73: "The experimental sites.." Mention these sites in a table if applicable or in the text.  

Line 80-83: this paragraph should be removed to the introduction section.

Line 87: provide the name of the company that made the drip irrigation system, size, and available specifications.

Line 98: please define the treatment clearly. It is ambiguous.

Line 129-138: define each equation's component and provide the proper unit.  

Results:

Overall, this section is very much written as a report, and the authors only reported the values that were in the figures without putting serious efforts into interrupting some of the findings scientifically. It is very well-known that DI has better water savings capability than BI. The authors' role is to interrupt their findings more deeply along with scientific facts than reported in this manuscript. They need to interrupt their findings from different aspects such as soil type and weather.   

Line 144: provide the soil water balance equation in M&M section.  

Line 164: “Due to more irrigation frequencies under DI treatment” Why more irrigation frequencies? Explain that.

Discussion:

This section is short, and I did not notice sufficient discussion of their findings with other related work. They must elaborate more and discuss their findings comprehensively.

Reviewer 2 Report

Due to the acreage of maize cultivation and the increasing problems with water supply, the subject of the article is very important. The results are all the more interesting as the efficiency of drop and border irrigation was compared in a large-scale experiment. The results obtained in this comparison are not surprising and confirm the available data from the literature. An interesting topic is the comparison of the results obtained in the described experiment, conducted over a large area, with the available results of plot experiments (small scale). Differences in potential benefits, depending on the scale of experience, are logical, but in my opinion, the authors did not use the opportunity to interpret the reasons for the obtained differences, depending on the scale of the research, and to have an interesting discussion. I believe that the work is at a very good level. The methodology, use of statistical tools, the way of presenting the results and conclusions do not raise any objections.

Detailed comments

It is good practice to avoid abbreviations in the Abstract. I understand that the abbreviations used do not raise doubts for a reader dealing with plant irrigation on a daily basis, but they do not have to be immediately readable for a reader dealing with, for example, plant physiology. Reading the text is easier when the abbreviation is always explained the first time you use a term (phrase). For example, the terms “drip irrigation” and “water use efficiency” were first used on page 1, l. 44,45 and there the explanation should be given in the brackets. The procedure for the following abbreviations  should be the same.

Reviewer 3 Report

The authors presented an interesting paper, and they focus on a topic related to the journal and important for the farmers. Although it is not a new topic, their 5 years of data show important results. The paper is well structured, but some sections need to be improved. To ensure that the paper reaches the quality expected for this journal following I have mentioned some issues that must be solved:

  • I suggest avoiding using in the keywords the same words used in the title (Water use efficiency).
  • Different citation methods are used in the paper; authors must uniformize them cites according to the journal's template.
  • At the end of the introduction, the authors must add a short paragraph detailing the aim of the paper and summarizing the efforts carried out to attain their data (methodology and difficulties). In this paragraph, the authors should highlight the importance of long-term data for these analyses.
  • In subsection 2.1 if possible, include a figure with the map of the region and indicate the suited area. The studied area is composed of a single plot or by several plots? Do they cover different solar exposure? Do they have the same rainfall? More information is required in the studied zone. More information about the used maize should be added.
  • I suggest changing the name of section 2.5. Probably, since in this section the authors are presenting the indicators they can label the subsection as Indicators. Section 2.6 and 2.5 can be merged.
  • Drip irrigation is well-known as a method to reduce water waste. Several papers show its benefits on maize as well as other crops. All this information must be analyzed in section 4. Moreover, the authors should include references about indicators for evaluating water efficiency, recent papers have been published about this in other crops, and there should be cites and included in their discussion:

Parra, L., Botella-Campos, M., Puerto, H., Roig-Merino, B., & Lloret, J. (2020). Evaluating Irrigation Efficiency with Performance Indicators: A Case Study of Citrus in the East of Spain. Agronomy10(9), 1359.

Fernández, J. E., Alcon, F., Diaz-Espejo, A., Hernandez-Santana, V., & Cuevas, M. V. (2020). Water use indicators and economic analysis for on-farm irrigation decision: A case study of a super high density olive tree orchard. Agricultural Water Management, 106074.

  • Authors must discuss the possible adverse effects of drip irrigation in the discussion such as the accumulation of salts and define if in their irrigation system do they include fertigation or not.
  • In general terms, section 4 must be extended, including the topics abovementioned and other ones, and more references must be added in this section.
  • Section 5 must be extended. Authors must include the future work linked to their results in an independent paragraph at the end of conclusions.

Round 2

Reviewer 1 Report

Comments to the Authors

The entitled manuscript of Effect of Drip Irrigation on Soil Water Balance and Water UseEfficiency of Maize in Northwest China has improved substantially after implanting the comments and suggestions. However, I do have a few suggestions before accepting the manuscript for publishing.

Line 31: remove the WUE and write the actual term (i.e., water use efficiency).

Line 142: equation (2), please, as I mentioned in my first-round review, define T, I, and P and provide each component's unit. Do the same for equations 3 and 4.   

Reviewer 3 Report

the authors have addressed all my comments

Author Response

We are very grateful for your constructive comments and suggestions. The comments provided are very careful and helpful, thus we should send our sincere thanks to all of you again!